# Do Invasive Mammal Eradications from Islands Support Climate Change Adaptation and Mitigation?

Peter J. Kappes [1],*, Cassandra E. Benkwitt [2], Dena R. Spatz [3], Coral A. Wolf [4], David J. Will [4] and Nick D. Holmes [5]

1   Coastal Research and Extension Center, Mississippi State University, Biloxi, MS 39532, USA
2   Lancaster Environment Centre, Lancaster University, Lancaster LA1 4YW, UK; c.benkwitt@lancaster.ac.uk
3   Pacific Rim Conservation, Honolulu, HI 96839, USA; dena@pacificrimconservation.org
4   Island Conservation, Santa Cruz, CA 95060, USA; coral.wolf@islandconservation.org (C.A.W.);
    david.will@islandconservation.org (D.J.W.)
5   The Nature Conservancy, Santa Cruz, CA 95060, USA; nick.holmes@tnc.org
*   Correspondence: pk565@msstate.edu

**Abstract:** Climate change represents a planetary emergency that is exacerbating the loss of native biodiversity. In response, efforts promoting climate change adaptation strategies that improve ecosystem resilience and/or mitigate climate impacts are paramount. Invasive Alien Species are a key threat to islands globally, where strategies such as preventing establishment (biosecurity), and eradication, especially invasive mammals, have proven effective for reducing native biodiversity loss and can also advance ecosystem resilience and create refugia for native species at risk from climate change. Furthermore, there is growing evidence that successful eradications may also contribute to mitigating climate change. Given the cross-sector potential for eradications to reduce climate impacts alongside native biodiversity conservation, we sought to understand when conservation managers and funders explicitly sought to use or fund the eradication of invasive mammals from islands to achieve positive climate outcomes. To provide context, we first summarized available literature of the synergistic relationship between invasive species and climate change, including case studies where invasive mammal eradications served to meet climate adaptation or mitigation solutions. Second, we conducted a systematic review of the literature and eradication-related conference proceedings to identify when these synergistic effects of climate and invasive species were explicitly addressed through eradication practices. Third, we reviewed projects from four large funding entities known to support climate change solutions and/or native biodiversity conservation efforts and identified when eradications were funded in a climate change context. The combined results of our case study summary paired with systematic reviews found that, although eradicating invasive mammals from islands is an effective climate adaptation strategy, island eradications are poorly represented within the climate change adaptation and mitigation funding framework. We believe this is a lost opportunity and encourage eradication practitioners and funders of climate change adaptation to leverage this extremely effective nature-based tool into positive conservation and climate resilience solutions.

**Keywords:** climate change funding; climate change strategies; climate adaptation; climate mitigation; climate resilience; conservation refugia; invasive alien species; island eradication; biosecurity; cross-sector funding; nature-based solutions

## 1. Introduction

Invasive Alien Species (hereafter, "IAS"; see Table 1 for definitions of key terms) are a primary driver of native biodiversity loss (Table 1) worldwide [1–3], with cascading effects on whole ecosystems and realized consequences to human societies and welfare [4–7]. Driven by globalization, the development of new commercial pathways and associated

increased movement of goods and humans around the globe, the frequency of IAS introductions and subsequent establishment continues to increase dramatically [8–11]. Nearly 40% of all first records of IAS were reported since 1970 [12] and by 2050 the number of established alien species per continent is predicted to increase by 36% [13].

**Table 1.** Definitions and references of key terms used in the paper that may not be commonly used or have may have alternate meanings among different scientific disciplines, practitioners, and/or climate funders.

| Term | Definition | Reference |
|---|---|---|
| Biodiversity loss | Loss of the variability among living organisms, including genetic (individual, subpopulation, and total population), species (uniqueness, abundances, and richness), functional (interactions and traits), and habitat diversity (different types and abiotic heterogeneity within them) from all sources, including terrestrial, marine, and aquatic ecosystems and ecological complexes of which they are part; for the purposes of our paper, this refers to loss of native biodiversity (see below) and not loss due to greater species richness resulting from the presence of IAS | [14,15] |
| Biosecurity | The actions needed to prevent, detect, and rapidly respond to the arrival of unwanted species, in a country (or island/archipelago) or between places within a country (or island/archipelago; with the common goal of protecting a country's (island's/archipelago's) economy, environment, and people's health from biological threats, such and plant and animal pests and diseases | [16] |
| Climate adaptation | Preparing for, coping with, or adjusting to climatic changes and their associated impacts | [17] |
| Climate change | Major shifts to the state of the climate (e.g., temperature, precipitation, and/or wind pattern) that occur over several decades or longer. May be due to natural processes or external anthropogenic changes to the composition of the atmosphere; in context of this paper change in climate is attributed directly or indirectly to human activity | [18] |
| Climate mitigation | Efforts to reduce or prevent emission of greenhouse gases and other anthropogenic climate forcing | |
| Climate resilience | The adaptive capacity for a socio-ecological system to: (1) absorb stresses and maintain function in the face of external stresses imposed upon it by climate change and (2) adapt, reorganize, and evolve into more desirable configurations that improve the sustainability of the system, leaving it better prepared for future climate change impacts | [19,20] |
| Conservation translocation | The deliberate movement of organisms from one site for release in another, with the intention to yield a measurable conservation benefit at the levels of a population, species, or ecosystem | [21] |
| Endemic species | Species that naturally occurs only in a single geographic area; in the context of this paper these species are located on single island/archipelago | [22] |
| Eradication | The complete and permanent removal of IAS | [23] |
| Invasive Alien Species (IAS) | Species that are either accidentally or intentionally introduced outside of their native range and have significant negative impacts on the native biodiversity, ecosystem services, and/or human well-being where they become established | [24] |
| Native biodiversity | Species that occur naturally in a given geographic area, as opposed to having been transported, inadvertently or purposefully, by humans | [25–27] |
| Propagule pressure | A composite measure of the introduction effort consisting of: the propagule size (i.e., the number of individuals introduced per introduction and the number/frequency of introduction events | [28] |
| Refugia | Areas that may facilitate the persistence of species during large-scale, long-term disruptive climatic change | [29] |

Climate change (Table 1) also contributes directly to native biodiversity loss (Table 1) [30–34], has already been directly linked to one known extinction [35,36], and is expected to accelerate in the future [37,38]. Range-restricted species (those that are unable to shift to alternate habitats with suitable climatic conditions or analog ecosystems) have the greatest extinction risk from changing climatic conditions [39,40]. This is particularly true for endemic species, which have an estimated 6% higher risk of extinction due to climate change than non-endemics [38]. Biodiversity hotspots, with their disproportionately high levels of endemic and range-restricted species, are forecast to be disproportionately impacted, with insular endemics most at risk of extinction [41–44].

Extinctions caused by IAS and/or climate change are particularly concerning for islands because islands harbor a disproportionate amount of the Earth's endemic species [42,45] and biodiversity [46,47]. Endemism richness estimates for plants and vertebrates are 9.5 and 8.1 times higher on islands than on the mainland [45]. In addition to being home to nearly 11% (~760 million people) of the world's population, 40% of globally threatened amphibians, birds, mammals, and reptiles [46,48] also inhabit islands. Island ecosystems are particularly susceptible to invasions [42,49,50] and IAS have disproportionate impacts on island ecosystems and native biodiversity compared to their impacts on continents [3,51]. Despite accounting for less about 5% of the Earth's land area, islands make up over 60% of the extinctions recorded on the planet [46]; the bulk of which (86%) were driven by IAS [1–3]. The loss of native biodiversity on islands is primarily due to predation impacts from invasive mammals, particularly rats, cats, and mongoose [1,52,53]. However, herbivory from rabbits, pigs, and goats also drives native biodiversity loss by dramatically altering and degrading island habitats [54–56]. The cascading effects from these impacts include alteration to both terrestrial and near-shore marine ecosystems [57–59]. This results in the loss of ecosystem function [60–62], which negatively impacts island economies and food security through crop damage, erosion, and native biodiversity loss, in addition to disease transmission to island wildlife and human populations [5–7]. Collectively these impacts make island ecosystems and economies less resilient (Table 1) to the impacts of climate change. Contrary to the well-known species-isolation relationship from island biogeography theory, where species richness decreases with isolation [63,64], the number of naturalized alien species on islands actually increases with isolation [65]. This unexpected relationship is hypothesized to be due to the reduced diversity and increased ecological naivete of native biota inhabiting more remote islands [65]. As such, IAS are and will continue to be a problem for even the most remote of oceanic islands, where their impacts are predicted to increase in association with climate change (see Section 2) [66].

The prevention control and eradication of IAS threats has become a foundational need for native biodiversity conservation on islands [15]. Preventing IAS from becoming established (biosecurity, Table 1) is the most tractable strategy and requires on-going comment. Deployed at scales from individual islands to island nations, it is increasingly necessary given global transport mechanisms [15]. Once established, IAS eradication (Table 1), particularly mammals, has proven to be an effective strategy for many islands around the globe, successfully eliminating threats in a discrete time frame, with demonstrable benefits to native species and ecosystems [62,67–69] and linkages to socioeconomic benefits for island communities [70]. To date, 88% of the more than 1500 known invasive vertebrate eradications undertaken on nearly 1000 islands worldwide were successful [71]. Historically, the motivation for eradicating IAS from islands was to conserve native biodiversity and promote ecosystem recovery [72]. Because the eradication of IAS from islands initially focused on native biodiversity conservation (the earliest attempts often focused on individual species) and now is implemented to facilitate ecosystem restoration, monitoring efforts have seldom investigated the possible contributions eradicating IAS from islands could make to climate resiliency and adaptation. As a result, the extent to which eradicating IAS from islands is being used as a potentially effective climate solution is currently unknown.

IAS are an important factor in the synergistic relationship between anthropogenic driven climate change and native biodiversity loss, two of the most pressing global envi-

ronmental crises of our time [30,73,74]. Climate change is expected to further the loss of native biodiversity through exacerbating global invasions and impacts from IAS [75–78], whereas IAS may affect the magnitude, rate, and impact of climate change, compounding their contributions to native biodiversity loss [79,80]. Here, we summarize the available literature that describes (1) the important characteristics that drive the positive feedback cycle between IAS, climate change, and native biodiversity loss, and (2) when eradications can likely contribute to improving resilience and adaptation to climate change or serve to mitigate climate impacts. We focus on eradications given project lifecycles have clear start and end dates and thus provide unique funding opportunities. However, the raison d'être we present here is relevant for other IAS strategies of control and biosecurity, where funding needs are on-going. We include case studies of the impacts of IAS and climate change on both terrestrial and near-shore marine ecosystems (Figure 1). To assess whether eradications are being considered or funded as a cross-sector conservation strategy, that addresses both climate change adaptation and native biodiversity loss objectives, we present the results of a review of the literature and a subset of global climate change funding awards. Although IAS include any species introduced through human aid outside its native range [24], we restrict our analysis and reviews to invasive vertebrates on islands, due to their known pervasiveness on islands and availability of eradication tools to successfully remove them from islands, resulting in high conservation gains [68,71]. Because mammals comprise the most common invasive vertebrate eradications, they are the primary focus of our case studies and examples. We therefore differentiate between the terms invasive vertebrates or mammals and the all-inclusive term IAS, which encompasses any invasive species, including non-vertebrates and plants. This synthesis aims to help decision-making by conservation practitioners and funders through filling an important knowledge gap of the synergistic relationships between invasive species and climate change on native biodiversity, and the potential for IAS eradications to improve climate change resilience and/or mitigation alongside biodiversity solutions.

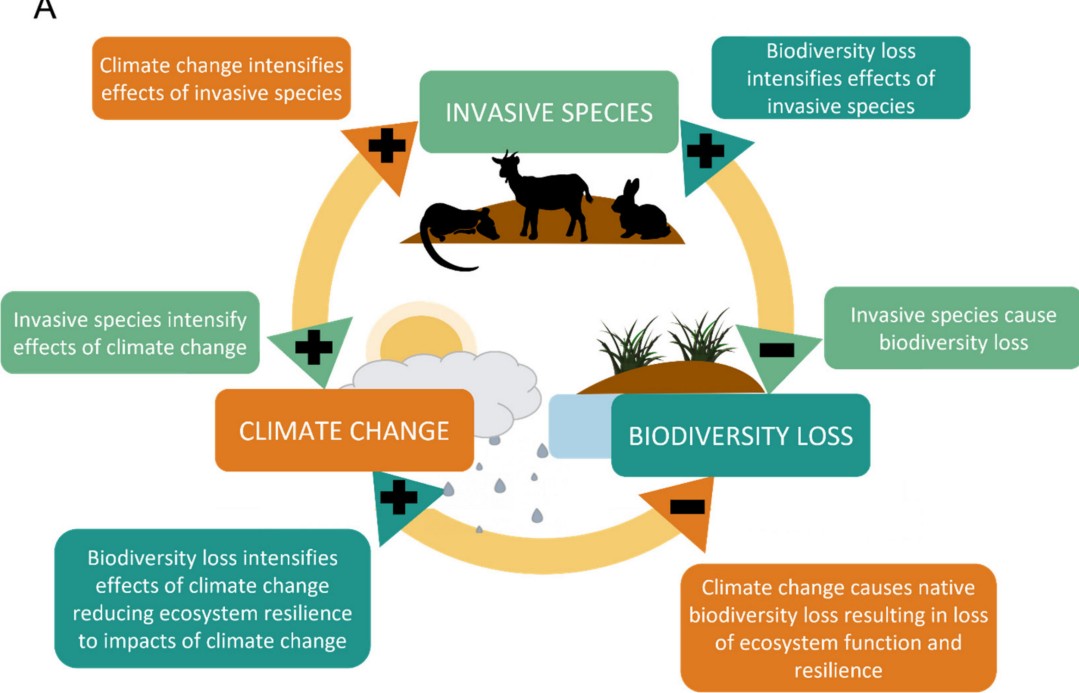

**Figure 1.** *Cont.*

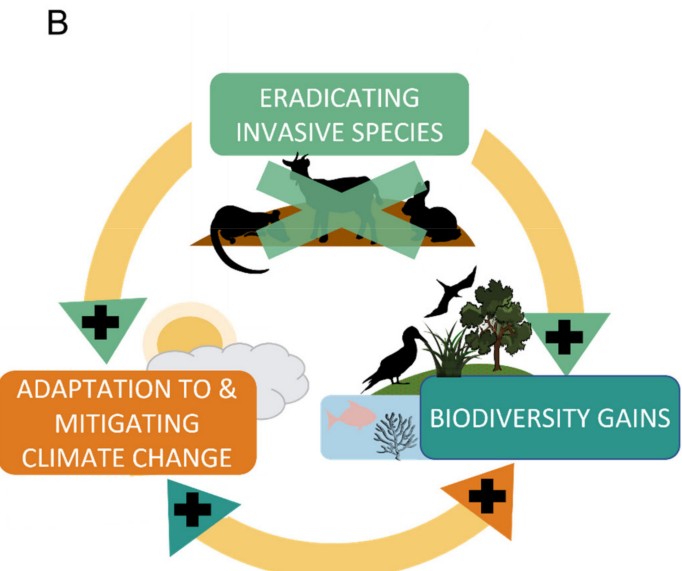

**Figure 1.** Conceptual summary of the major links among climate change, invasive alien species, and native biodiversity. Arrows indicate direction of influence. Arrowheads are color-coded to match the driver of influence. Symbols in arrowheads indicate if the relationship between influencer positively (+) or negatively (–) influences recipient box. (**A**) Illustrates the synergies between climate change, IAS, and native biodiversity loss and how they form a positive feedback cycle. Climate change intensifies the effects of invasive alien species (IAS), for example by increasing the risk of spread and establishment, increasing population sizes, and/or altering their diet or behavior; while also directly impacting native biodiversity via creating mismatches in species niche requirements, sea-level rise, and extreme weather events. IAS intensify and reduce the resilience of ecosystems to climate change by reducing native biodiversity, for example by increasing susceptibility of native habitats to climate change impacts such as erosion and marine heat waves, and/or IAS increase atmospheric carbon by causing a loss of carbon sequestration provided by native vegetation and undisturbed soil. In turn, native biodiversity loss is driven by compounded impacts of climate change and IAS, which degrade ecosystems making them more vulnerable to both climate change and IAS. (**B**) Illustrates how eradicating IAS from islands breaks the positive feedback cycle, improving ecosystem resilience to and mitigating the impacts of climate change while protecting native biodiversity and promoting recovery. (Figure adapted from the Invasive Species Centre, available online: https://www.invasivespeciescentre.ca/invasive-species/what-is-at-risk/climate-change/ (accessed on 17 July 2021).

## 2. Climate Change, IAS, and Native Biodiversity Loss: The Positive Feedback Loop

In general, climate change and IAS are predicted to interact positively [41,76]. Because most IAS are highly adaptable generalists that thrive in disturbed systems, they are expected to adapt more readily to changing climatic conditions and outcompete range-restricted species and/or species with more specialized requirements [75,81], including on islands [41]. Although climate change may reduce propagule pressure (Table 1) for some species [82], an important factor of invasion success, this is likely to be offset by other synergistic interactions between climate change and IAS (Table 2). New invasion pathways for IAS, allow access to previously inaccessible habitats and/or shortened transit times that may increase the likelihood of survival from source populations to "closer" destinations (e.g., opening of Northwest Passage with loss of sea ice) [77,83,84]. Altered and more frequent extreme weather events expected with climate change [85] may further assist the dispersal of IAS [86–88]. Warmer temperatures and altered precipitation patterns associated with climate change will influence both the thermal and resource restrictions of some IAS, facilitating the range expansions or population increases of some invasive species [41,83,89]. For example, on the sub-Antarctic Marion Island, a 430.0% increase in the House mouse (*Mus musculus*) density was related to the number of increasing precipitation-free days over the past 30 years [90]. The urgency of the recently successful rodent eradication on South Georgia was in part predicated on the potential spread of rodents between bays and peninsulas separated by glaciers that will likely retreat with warmer temperatures [91].

Climate change is also expected to intensify the impacts of IAS. For example, in Australia, feral cats have a greater impact on native biodiversity in drier habitats, which are expected to expand with lower projected rainfall [92]. In the Canary archipelago, invasive European rabbit (*Oryctolagus cuniculus*) populations are projected to increase due to increased temperatures and reduced precipitation [93]. Increased prey resources combined with increasing temperatures are likely to enhance rodent survivorship and reduce metabolic costs of thermoregulation, allowing rodents to divert more resources to reproduction [94,95], increasing the magnitude of their impacts [90]. Invasive species may also shift their diets in response to altered temperature and precipitation patterns [90,96,97]. In some cases, this increased damage to native species and ecosystems will be accompanied by additional socio-economic costs. For example, rodent outbreaks triggered by increased food availability following extreme weather events threaten agricultural production and food security in many parts of southeast Asia and Australia [98,99].

Even in the absence of IAS climate change directly impacts native biodiversity by creating mismatches in species niche requirements, sea-level rise, and extreme weather events [75]. However, the synergistic relationship between IAS and climate change compounds their contributions to native biodiversity loss and magnifies their social and economic impacts (Figure 1) [75]. As ecosystems lose native biodiversity, important ecosystem services are degraded and resilience to climate change impacts and potential mitigating influences are compromised [100–102]. In addition, degraded ecosystems are more susceptible to the establishment of IAS and subsequent impacts that further native biodiversity loss [1,3], completing the destructive positive feedback loop. Although the interplay between climate change, IAS, and native biodiversity loss is becoming more well-known, we acknowledge that there is still much to learn about the synergistic relationship between them, particularly climate change and IAS [84,103].

**Table 2.** Examples summarizing key aspects of the synergistic interaction between climate change and invasive alien species (IAS) forming part of the positive feedback loop between climate change, IAS, and native biodiversity loss. "Focal IAS" describes the invasive taxa examined. "Geography/Location" refers to the geographic focus or location of the study. "Impact" describes the outcome of the interaction, such that "Exacerbates" indicates the outcome benefits IAS or exacerbates climate change and "Reduces" indicates the outcome is detrimental to IAS or mitigates climate change. "Determined" indicates if the outcome was predicted and/or observed. The table is not exhaustive.

| Synergistic Interaction | Focal IAS | Geography/Location | Impact | Determined | Reference |
|---|---|---|---|---|---|
| Review or summary papers documenting more than one type of synergistic interaction * | | | | | |
| | Vertebrates, invertebrates, and plants | Global | Exacerbates and reduces IAS | Predicted and observed | [41,75,77,79, 80,83] |
| | | Australia's World Heritage Sites | Exacerbates IAS | Predicted and observed | [81] |
| | | Antarctica and Southern Ocean | Exacerbates IAS | Predicted and observed | [84,104] |
| | | Australia | Exacerbates IAS | Predicted | [92] |
| | | Great Britain | Exacerbates and reduces IAS | Predicted | [105] |
| | Plants, invertebrates, fishes, and birds | Global | Exacerbates and reduces IAS | Predicted | [78] |
| | Fish | Freshwater ecosystems | Exacerbates and reduces IAS | Predicted | [106] |
| | Top 100 most invasive species [107] | Global (including islands) | Exacerbates and reduces IAS | Predicted | [76] |

**Table 2.** *Cont.*

| Synergistic Interaction | Focal IAS | Geography/Location | Impact | Determined | Reference |
|---|---|---|---|---|---|
| Climate change alters transport and introduction of IAS | | | | | |
| Tourism and new trade routes | Vertebrates, invertebrates, and plants | Global | Exacerbates IAS | Predicted and observed | [12] |
| Extreme weather events | African locusts (*Schistocerca gregaria*) | African windward islands | Exacerbates IAS | Observed | [108] |
| | Plants | Coastal wetlands | Exacerbates IAS | Predicted | [109] |
| | Green iguana (*Iguana iguana*) | Anguilla | Exacerbates IAS | Observed | [86] |
| | Birds | Global | Exacerbates IAS | Observed | [88] |
| Climate change alters distribution of existing IAS | | | | | |
| | Vertebrates, invertebrates, and plants | Global | Exacerbates IAS | Predicted and observed | [110,111] |
| | Norway rats (*Rattus norvegicus*) | Australia | Exacerbates IAS | Predicted | [92] |
| | | South Georgia Island | Exacerbates IAS | Predicted | [91] |
| Altered climatic constraints | | | | | |
| Creation of suitable habitat | Vertebrates, invertebrates, and plants | Global | Exacerbates IAS | Predicted and observed | [8,112] |
| | Gastropods | Global | Exacerbates IAS | Predicted | [11] |
| | European rabbit | Tenerife, Canary Islands, Spain | Exacerbates IAS | Predicted | [93] |
| Removal of barriers preventing IAS from establishing population | Norway rats | South Georgia Island | Exacerbates IAS | Predicted | [91] |
| Native species become invasive under altered climatic conditions | Vertebrates, invertebrates, and plants | Global | Exacerbates IAS | Observed | [113] |
| | Mountain pine beetle (*Dendroctonus ponderosae*) | North American forests | Exacerbates IAS | Observed | [114] |
| Extreme weather events | Rats (*Rattus* spp.) | Southeast Asia | Exacerbates IAS | Predicted and observed | [99] |
| | | Myanmar | Exacerbates IAS | Predicted and observed | [98] |
| Reduction in propagule pressure | Insects | Global | Exacerbates and reduces IAS | Predicted | [82] |
| Climate change alters impacts of existing IAS | | | | | |
| | Vertebrates, invertebrates, and plants | Global | Exacerbates IAS | Predicted and observed | [111] |
| | House mouse | Sub-Antarctic Marion Island, South Africa | Exacerbates IAS | Observed | [90] |
| | European rabbit | Tenerife, Canary Islands, Spain | Exacerbates IAS | Predicted | [93] |

**Table 2.** *Cont.*

| Synergistic Interaction | Focal IAS | Geography/Location | Impact | Determined | Reference |
|---|---|---|---|---|---|
| IAS outcompete native species under climate change | *Maesopsis eminii* | East Usambara mountain forests, Tanzania | Exacerbates IAS | Observed | [115] |
| | Black rat (*R. rattus*) | Santiago Island, Galápagos Islands, Ecuador | Exacerbates IAS | Predicted | [97] |
| Behavioral change | Feral cats (*Felis catus*) | San Clemente Island, California | Exacerbates IAS | Predicted and observed | [96] |
| Climate change alters effectiveness of IAS control and recovery of native biodiversity | | | | | |
| Seasonal limitations | European rabbit | Aotearoa/New Zealand | Exacerbates IAS | Observed and predicted | [116] |
| | | Kerguelen archipelago, France | Exacerbates IAS | Observed | [117] |
| IAS exacerbate climate change | | | | | |
| Reduction of carbon sequestration and release of carbon | Invertebrates and plants | Global | Exacerbates climate change | Predicted | [75] |
| | Feral pigs | Global | Exacerbates climate change | Observed | [118] |
| | Mountain pine beetle | North American forests | Exacerbates climate change | Observed | [119] |
| | Rat spp. (*Rattus* spp.) | Aotearoa/New Zealand | Reduces climate change | Observed | [120] |
| Destabilize coastal wetlands and anthropogenic flood-control structures | Nutria (*Myocastor coypus*) | Global | Exacerbates climate change | Predicted | [79] |

\* Review papers highlight that impacts of climate change will vary spatially and temporally, and that IAS will respond differently, which will mean that outcomes are predicted to be mixed depending on the combination of spatial, temporal, and IAS being considered. We direct readers to these references for more detailed information and case studies not summarized here.

## 3. Evidence for Invasive Species Eradications as a Climate Adaptation Strategy

IAS eradications address two of the seven climate adaptation strategies identified by the U.S. Climate Change Science Program [121], by assisting island communities to prepare for, cope with, and adjust to climatic changes and their associated impacts [17]. These are (1) to promote ecosystem resilience, which involves reducing anthropogenic stresses (i.e., the "Safe operating space" argument [122]), to preserve or enhance the resilience of ecosystems to regional, uncontrollable climate stresses, and (2) the creation or maintenance of refugia, particularly with the potential for relocation and restoration of species threatened by climate change elsewhere. In the following section, we briefly summarize the evidence supporting invasive mammal eradications as an effective climate adaptation strategy for achieving these goals beyond their already well-established role in conserving native biodiversity.

### 3.1. Invasive Mammal Eradications and Ecosystem Resilience

Eradicating invasive mammals from islands removes a major extinction stressor, subsequently allowing island species to become more resilient to the stresses imposed by climate change (Table 3). For example, removing invasive mammals confers resiliency to native populations via removal of competitors and/or predators, improving population

attributes such as size, density, and growth rates [123]. Removing IAS can also improve both habitat quality and restore food web and trophic dynamics in terrestrial [62,124–126] and marine environments [57,127–129], important components in a healthy ecosystem that is resilient to climate change [130].

In terrestrial systems, invasive herbivore eradications can reduce erosion rates, and increase plant production that aids in stabilizing soils [131–135]. On subantarctic Macquarie Island (southwestern Pacific Ocean), the impacts of invasive rabbits and climate change interacted to negatively impact the island ecosystem in a variety of ways [136]. Increased temperatures, annual rainfall, and wind speeds were additive to the impacts of invasive rabbits and accounted for the cascading impacts of severe vegetation loss, increased rates of slope erosion, increased frequency of landslides and decreased albatross breeding success.

**Table 3.** Summary of evidence provided in the text that eradicating invasive mammals from islands may benefit climate change adaptation/resilience to or mitigation of climate change. This includes studies comparing islands with and without IAS. "Eradication" refers to the process promoted by eradicating invasive mammals. "Focal Taxa" refers to the taxa eradicated or compared with and without. "Location/Geography" refers to the study site. "Outcome" refers to the way in which the eradication or absence of IAS contributes to a climate adaptation/resiliency or mitigation outcome.

| Eradication | Focal Taxa | Location/Geography | Outcome | References |
|---|---|---|---|---|
| *Enhance species/ecosystem resilience* | | | | |
| Improve native population (e.g., size, density, growth rate) or recruitment/rediscovery of extirpated native spp. (e.g., plants, seabirds) | Mammals | Global Islands | Positive: increases in population attributes and/or recruitment of new and/or extirpated species | [68] * |
| | | | Positive: recovery of impacted populations | [54] * |
| | | | Positive: increased population growth rate, nesting success, and enhanced adult survival | [69] * |
| | Mammals | Aotearoa/New Zealand | Positive: recovery of numerous native plants, invertebrates, and >70 spp. of terrestrial vertebrates | [67,137] * |
| | Black rat (*R. rattus*) | Palmyra Atoll, Line Islands Central Pacific Ocean | Positive: increased recruitment of six native tree spp.; important for species dependent on these speciesNegative: increased recruitment of non-native tree spp. | [62] |
| Create refugia | Invasive mammalian predators | Midway Atoll | Positive: creation of predator-free areas to recruit/translocate species at risk from sea-level rise | [138] |
| | | Lehua Island, Hawai'i, USA | Positive: creation of predator-free areas to recruit/translocate species at risk from sea-level rise | [139–142] |
| Restore food web and trophic dynamics, habitat quality | Rat (*Rattus* spp.) and feral cat (*Felix catus*) | Mercury Archipelago, Aotearoa/New Zealand | Positive: more diverse macroalgae communities at islands longer post eradication and more like never invaded islands | [128] |

**Table 3.** *Cont.*

| Eradication | Focal Taxa | Location/Geography | Outcome | References |
|---|---|---|---|---|
| | Mammals | Vanua Levu, Fiji | Positive: seabird-derived nutrient subsidies enhance growth of dominant reef-building spp. | [127] |
| | Rat spp. (*Rattus* spp.) | Aotearoa/New Zealand | Positive: seabird burrow density mediated soil and vegetation dynamics and were slightly higher on islands post eradication | [124] |
| | Norway rat (*R. norvegicus*) | Tromelin Island, Western Indian Ocean, France | Positive: increase in breeding pairs of two seabird species and recruitment of two seabird species; increase in vegetation coverNegative: increase in invasive House mouse (*M. musculus*) | [126] |
| | | Hawadax Island, Aleutian Islands, Alaska | Positive: recovery of terrestrial birds, shorebirds, and recolonization by seabirds; community-level recovery: return of three-level trophic cascade in rocky intertidal, with decreases in invertebrate species and increases in fleshy algal cover | [143] |
| | Black rat | Chagos Archipelago, Indian Ocean | Positive: seabird-derived subsidies enhance coral reef productivity and functioning and may increase resilience of reefs to climate change | [57,60,144, 145] |
| Reduce erosion, stabilize soils | Feral pig (*Sus scrofa*) | Isla del Coco, Costa Rica | Positive: vegetation recovery and reduction in erosion | [132] |
| | | O'ahu, Hawai'i, USA | Positive: Runoff volume was lower from plots excluding pigs | [135] |
| | Feral sheep (*Ovis aries*) and cattle (*Bos primigenius*) | Santa Cruz Island, California | Positive: transition from grass-dominated systems to woody systems, increased woody vegetation and overstory | [133] |
| | Feral goat (*Capra hircus*) | Guadalupe Island, Mexico | Positive: recovery of native vegetation and rediscovery of species thought extinct or extirpated | [134] |
| | European rabbit (*O. cuniculus*) | Macquarie Island, Australia | Positive: recovery of native vegetation providing high-quality nesting habitat for three species of albatross, increased soil stabilization resulting in reduced erosion and increased reproductive success of albatrosses | [136] |

**Table 3.** *Cont.*

| Eradication | Focal Taxa | Location/Geography | Outcome | References |
|---|---|---|---|---|
| | | Kerguelen archipelago, France | Positive: combination of rabbits and climate change decimated plant cover and increased erosion; following eradication increased plant richness and reduction in erosionNegative: increased plant richness following eradication driven by invasive plant species adapted to warmer, drier climates due to climate change has resulted in an increase in soil erosion particularly where rabbits are still present | [117,146, 147] |
| *Mitigate climate change* | | | | |
| Increased carbon sequestration | Feral sheep and cattle | Santa Cruz Island, California | Positive: transition from grass-dominated systems to woody systems, increased woody vegetation and overstory result in 70% and 17% increase in stored carbon and nitrogen | [133] |
| | *Rat* spp. | Aotearoa/New Zealand | Negative: islands with rats had higher rates of carbon sequestration | [120] |
| | Herbivores | | Positive: removal resulted in increased carbon sequestration | [148] |
| | European rabbits | Australia | Positive: removal of rabbits could be more effective way to sequester carbon than planting trees | [149] |

* Table is not exhaustive. We direct readers to review papers that contain further references with more detailed information and case studies not summarized here.

Following the removal of invasive rabbits, vegetation recovered rapidly, providing high-quality nesting habitat (e.g., protection from extreme weather such as high winds and heavy rain), increased soil stability with reduced erosion (e.g., protection from increased rainfall), and early signs of increases in albatross breeding reproduction. In the subantarctic Iles Kerguelen (south Indian Ocean) archipelago, the introduction and subsequent increase in abundance of rabbits was followed by a decline of the dominant native plant species and a dramatic increase in erosion [146]. The combination of rabbit-herbivory and changing climate conditions (e.g., droughts) cumulatively contributed to the greatest declines in plant cover and increased erosion [147]. Following the removal of rabbits, low rainfall appeared to be the primary factor inhibiting native plant regeneration and, although significant increases in plant species richness occurred, it was largely driven by introduced plant species better adapted to warmer, drier climates [117,147]. Following a successful eradication, the legacy effects of invasive species may hamper the ability of ecosystems to naturally recover at the pace needed to become resilient to climate change [150]. Thus, subsequent management (e.g., reforestation, erosion control, restoring seabird populations) is an important restoration component following successful eradication [134,151,152].

For marine systems, a reduction in terrestrial erosion due to the removal of invasive mammals can also lead to a reduction of sedimentation on coral reefs, which can help ameliorate the effects of climate change on corals by improving water quality [153,154]. The removal of rats, a pervasive invasive omnivore on islands, was also shown to play a role in the resilience of nearshore marine systems, including in temperate and tropical latitudes [128,129,144]. Islands free of introduced predators have larger breeding seabird

populations than nearby invaded islands, and these seabirds transport large amounts of nutrients from their feeding grounds in the open ocean to terrestrial and coastal systems [145,155,156]. In coral-reef ecosystems, these nutrients are taken up by a variety of organisms [57,127,157,158] and lead to higher fish productivity, biomass, and ecosystem functioning [57,60]. By comparing islands with introduced rats (and few seabirds) to nearby rat-free islands with abundant seabirds before versus after an extreme marine heatwave, Benkwitt, Wilson and Graham [144] demonstrated that seabird-derived nutrient inputs also altered the response of coral reefs to climate change. Although the coral loss was similar between rat-free and rat-infested islands, herbivorous fish biomass and cover of crustose coralline algae were higher around rat-free islands, both of which may speed reef recovery. Corals also grow faster near islands with abundant seabirds [127], which is likely to further enhance recovery rates. Finally, lab-based studies have shown that biological nutrients with ratios of nitrogen to phosphorous such as those in seabird-derived nutrients may increase the thermal tolerance of corals [159,160], suggesting there may be additional benefits of seabirds for reefs during less severe heat waves. Combined with the fact that eradicating invasive rats can restore nutrient subsidies from seabirds to both islands and coral reefs [145], these studies suggest that eradications may help promote coral reef resilience to climate change.

Importantly, healthy marine structures such as corals, coastal wetlands, and seagrass beds help to mitigate the severity of coastal damage during extreme weather events [79,161,162], which are expected to increase in frequency and intensity with climate change [18,85]. Removing IAS also has carry-on effects on human livelihoods, including tourism, food security, and fisheries revenue. High marine biomass increases food security and fisheries revenue [162]. Associated improvements in water quality, including intact terrestrial and marine floral and faunal communities, can help to attract tourists with associated economic benefits. Thus, IAS eradications can simultaneously promote sustainable food production and consumption systems, improve human health and water quality, generate employment and opportunities for climate change mitigation [70].

### 3.2. Invasive Mammal Eradication and the Creation of Refugia

Refugia provide protected areas from which species can persist and are particularly important as climate change alters habitat quality and availability via changes in sea level and rainfall [29]. The eradication of invasive mammals from islands is critical for species that will seek refugia or will require assistance to reach refugia via conservation translocation (Table 1) [141,163]. The creation of refugia will be particularly important on high elevation islands as low-lying islands become inundated from increased sea levels and flooding from extreme weather events (Table 3). By one estimate, up to 12% (152) of the French-administered islands worldwide could be submerged by 2100 [164]. Another study estimated that a 1 m rise in sea level by 2100 would entirely submerge 6% of the 4447 islands comprising the top ten global insular hotspots of native biodiversity [43]. The complete submersion of many of these islands would result in total loss of terrestrial habitat and associated catastrophic native biodiversity loss [33,43,139,164–166]. Forty-three percent (42 species) of all threatened seabirds occur on at least one island with a medium to high risk of inundation [167]. Furthermore, the inland displacement of people from vulnerable coastal areas increases human-wildlife conflict and could further push coastal species into sub-optimal habitats, adding to extinction risk if not directly causing their extinction [168]. This highlights the need for careful planning of eradications so that limited resources are not committed to islands that may be inundated, although in some instances eradications on islands at high risk of submersion may serve as a means to prevent imminent extinction and provide extra time for eradications on higher elevation islands and/or relocation efforts to be implemented elsewhere [33,167]. For example, the eradication of black rats (*R. rattus*) from low-lying Midway Atoll in 1996 [138] created temporary refugia for Black-footed Albatross (*Phoebastria nigripes*), whose populations are now increasing. Subsequent

action to create new colonies via the translocation of albatrosses to high elevation islands is possible thanks to such early invasive species eradication efforts [141,142].

Lehua Island (213 m elevation; 110 ha), an uninhabited island 31 km west of Kaua'i Island in the main islands of the Hawaiian Archipelago, once provided habitat for at least 18 native seabird species [140]. The introduction of invasive rats (*R. exulans*) and rabbits led to the extirpation of some of these seabird species and depressed the populations of others. Many of these species were also impacted by invasive mammals at other breeding colonies in the Hawaiian Islands, including those on high elevation islands in the main Hawaiian Islands [169]. As a result, most of these seabird species nest on the low elevation Northwestern Hawaiian Islands where invasive mammals were absent but the risk of inundation due to extreme weather events and sea-level rise is high [139,170]. This risk was exemplified recently when the 4.5 ha East Island of the French Frigate Shoals – a major green sea turtle (*Chelonia mydas*) and seabird nesting area – was wiped off the map when it became completely eroded and submerged overnight by a hurricane. Thus, the plan to eradicate invasive mammals from the relatively high elevation Lehua Island was in large part due to the opportunity to protect seabirds from the dual impacts of invasives and climate change across the Hawaiian Archipelago [139,140]. Now free of invasive mammals, Lehua Island provides safe seabird nesting habitat from both sea-level rise and invasive predators found on the main islands, and planning for the next phases of the project—social attraction to reintroduce native seabird species—is underway (pers. comm. P. Baiao; Table 3). Indeed, the application of eradication tools in conjunction with reintroductions and translocations to high elevation refugia is an important aspect of seabird conservation across the region (e.g., Kaho'olawe Island eradication project, the "No Net Loss" initiative) [139,171,172].

### 3.3. Invasive Mammal Eradication and Climate Mitigation

In addition to the potential of invasive mammal eradications for enhancing climate adaptation and resilience, there is also evidence from multiple ecosystems that removing invasive mammals can help mitigate climate change (Tables 1 and 3) by promoting carbon sequestration [70,146]. Estimates of invasive mammal impacts on terrestrial carbon sequestration include a recent study's calculation that by uprooting soil, feral pigs (*Sus scrofa*), most of which are invasive, release 4.9 million metric tons (MMT) of $CO_2$ per year globally (equivalent to 1.1 million passenger vehicles) [118]. Yet another study on removing rabbits from Australia proposes that carbon sequestration resulting from their removal could be more cost-effective than planting trees [149]. However, in the case of rabbit removal and invasive mammals in general, the magnitude of carbon sequestration has not been established, and case studies calculating these carbon benefits are few. On Santa Cruz Island, Channel Islands, California, researchers found evidence for increased carbon stores 30 years following the eradication of introduced, feral ungulates (sheep (*Ovis aries*) and cattle (*Bos primigenius*)) that had heavily impacted the native plant community and caused erosion [173,174]. Native woody vegetation replaced grasses and bare ground, resulting in increases of 97% and 17% total above- and belowground carbon storage and nitrogen pools, respectively [133]). Controlling invasive herbivores in New Zealand (e.g., deer (7 species), feral goats (*Capra hircus*), and brushtail possums (*Trichosurus vulpecula*)), also resulted in positive carbon gains. However, this occurred primarily through complex indirect mechanisms that are likely to be localized to areas of highly palatable early-successional vegetation and high herbivore densities where control initiates the rapid development of woody vegetation [148].

In contrast, at another site in New Zealand, carbon sequestration was higher in the presence of invasive species, with rat-invaded islands having carbon gains compared to rat-free islands inhabited by healthy seabird populations that modify the habitat and remove understory vegetation [120]. Thus, it cannot be assumed that all removals of invasive species will result in terrestrial carbon gains [125]. An expanded understanding of when (i.e., under what circumstances) the removal of introduced species can not only result in the long-term recovery of island plant communities but may also mitigate the effects of

anthropogenic greenhouse gases is worthy of further study. In addition, there has been a growing recognition of the importance of "blue carbon"–carbon naturally stored in coastal and marine ecosystems [175,176]. Links between invasive terrestrial island mammals and coastal ecosystems that are rich in carbon (e.g., mangroves, seagrasses) are expected, however, further research is needed to better define these pathways and evaluate the indirect impacts that invasive terrestrial mammals have on blue carbon sequestration.

## 4. Systematic Review of the Literature and Funding

### 4.1. Literature

To quantify whether the eradication of invasive vertebrates on islands is being considered as a strategy to address climate change, and to summarize evidence of the contributions of invasive mammal eradications to climate change adaptation and mitigation, we first searched Web of Science (WoS) for published papers. We used the topic search words: (invasive OR introduced or alien) AND (vertebrate OR mammal OR herbivore OR predator OR rodent OR ungulate OR cat OR rabbit OR hare OR goat OR sheep OR mouse OR rat OR stoat OR possum OR fox OR pig OR mongoose) AND (eradicat* OR removal) AND (climate). We selected these keywords to identify papers focused on both invasive vertebrate eradication and climate change. We did not include "island" as a search term because preliminary searches revealed this was too limiting, and instead we filtered for studies focused on islands during our secondary screening. We specifically listed invasive mammals in our search terms based on the most common invasive vertebrates on islands from the Database of Island Species Eradication (DIISE) [71]. To further determine whether eradication projects explicitly consider climate change, we searched the past 10 years of proceedings from the most widely recognized conferences on vertebrate pests and island eradications (Vertebrate Pest Conference (2012, 2014, 2016, 2018, 2020), Australasian Vertebrate Pest Conference (2011, 2014, 2017) and the International Conference on Eradication of Island Invasives (2011, 2017). For all proceedings, we searched for the term "climate" in the titles, abstracts, and keywords. We screened all papers returned from both the WoS and proceedings searches to determine the type of paper (review/commentary or empirical/modeling) and ensure that the following criteria were met: focus on an invasive vertebrate, focus on islands, focus on climate change (climate change or related term appears in the title, abstract, or author keywords), and focus on invasive species eradication actions (eradication, removal or related management term appears in the title, abstract, or author keywords).

Overall, we found very few examples of published studies or conference presentations that explicitly addressed both invasive mammal eradications on islands and climate change (Tables S1 and S2). For example, our WoS search returned only 130 results, of which only 24 (14 empirical/modeling studies and 10 review/commentary papers) were relevant based on the secondary screening. By contrast, a WoS search with our same keywords minus any climate-related terms returned 3355 results. Similarly, only 33 abstracts from recent conferences focusing on invasive species mention "climate", and only eight of these passed the secondary screening (three empirical/modeling studies and five review/commentary papers).

Despite the relatively small number of studies (32) obtained from the systematic literature review, they still offer insights into the current state of knowledge regarding the integration of research on invasive vertebrates and climate change. Papers focused primarily on (1) the effects of climate change versus invasive mammals on native species and ecosystems (*n* = 15 papers) and/or (2) the influence of climate change on the spread, establishment, impacts, or management of invasive species (*n* = 16 papers). There were few examples of invasive mammal eradications as a potential climate solution beyond its well-established role in enhancing the population size and biodiversity of native species, yet these cases highlight that eradications may improve climate change adaptation and mitigation through diverse pathways and in diverse ecosystems. For example, in marine systems, seabird-derived nutrients, which are only present in large quantities around is-

lands that lack invasive rats, may improve recovery of tropical coral reefs following marine heatwaves [144] (see Section 3.1). In terrestrial systems, invasive herbivore eradication is linked to reduced erosion rates and the regeneration of native vegetation that can enhance carbon sequestration [70,146] (see Section 3.3). Finally, several studies highlight the potential of invasive mammal eradications to provide new refugia for threatened species such as seabirds, particularly if islands are prioritized for eradications that are less susceptible to future sea-level rise [140,167,177,178] (see Section 3.2).

Ultimately, although there are multiple logical links between IAS eradications and climate change [70], in most systems the benefits of eradications for increasing resilience to and mitigating the effects of climate change remain to be rigorously tested. Indeed, even amongst papers returned with our search terms of "eradicat*" or "removal", there were very few investigations of islands where invasive species had been eradicated [117,179]. Instead, eradication or removal of invasive species was typically mentioned as recommendations for future management action. Thus, empirical studies that quantify the effects of eradication on climate change resilience and adaptation are urgently needed. Nonetheless, even the small number of empirical articles published provide compelling evidence of the linkages between invasive mammal eradication and improving climate solutions, warranting further attention by practitioners and decision-makers to consider climate change in eradication planning and decision-making.

### 4.2. Funding Sources: Awarded Projects

We determined whether funders are supporting projects that propose to mitigate climate change and/or promote ecosystem adaptation or resilience to the impacts of climate change in connection with invasive vertebrate eradications. To do this we reviewed the 6729 funded projects from the websites of the following four large entities we thought were most likely to support global climate initiatives: (1) Global Environment Facility (GEF), the operating entity of the financing mechanism to the United Nations Framework Convention on Climate Change (UNFCCC) ($n$ = 5288; https://www.thegef.org/ (accessed on 1 October 2021)), (2) the Darwin Initiative ($n$ = 1164; https://www.gov.uk/government/groups/the-darwin-initiative (accessed on 1 October 2021)), (3) the Green Climate Fund ($n$ = 173; https://www.greenclimate.fund/ (accessed on 1 October 2021)), and (4) the Wildlife Conservation Society's Climate Adaptation Fund ($n$ = 104; https://www.wcsclimateadaptationfund.org/ (accessed on 1 October 2021)). Each website had a different search feature for identifying projects, thus, the initial search used the most relevant search terms available to allow us to screen for relevant project topics (see Table 4). After each search result we screened the titles and/or abstracts of these projects using the following terms: climate*, invasive, introduced, alien, eradicate*, removal, and/or IAS. For any project meeting the above criteria, we conducted a secondary screening of the project proposals, summaries, and/or reports to ensure that the project focused on invasive vertebrates, island(s), and climate change (where "climate change", "climate change mitigation", "climate adaptation", "climate mitigation", or some combination appeared in the title, abstract, or project keywords), and that at least one objective of the project was the eradication of invasive vertebrates from an island(s) to help mitigate climate change and/or improve ecological adaption or resilience to the impacts of climate change.

Of the 6729 projects, only 79 projects met the minimum criteria for inclusion in the secondary screening process. Of these 79, only 9 projects (11%) proposed to eradicate invasive vertebrates from islands to help mitigate and/or promote adaptation or resilience to the impacts of climate change (Tables 3 and S3), in addition to benefits to native biodiversity. Although most of the funded projects did not meet our specific search criteria, nearly a quarter of the 79 projects in the first screening suggested interactions between invasive species and climate change and/or the effects of climate change on ecosystem resilience could influence project objectives and outcomes. Additionally, a quarter of the 79 projects that did not identify eradicating IAS as an objective did include IAS management (e.g., building capacity to identify IAS prior to their establishment, increasing and

improving efforts and techniques to prevent the introduction and establishment of IAS (i.e., biosecurity)), highlighting the recognized importance of IAS prevention and management that will inherently benefit native biodiversity, which would naturally improve ecosystem adaptation and resilience to the impacts of climate change.

**Table 4.** Summary of the number of projects funded by Global Environment Facility (GEF), Darwin Initiative (DI), Green Climate Fund (GCF), and the Wildlife Conservation Society's Climate Adaptation Fund (WCS) as they went through our review process to identify projects that explicitly proposed to eradicate an invasive mammal from an island(s) and stated part of the reasoning behind requesting funding was to improve resilience to the impacts of climate change.

| Funding Entity | GEF | DI | GCF | WCS | Total |
|---|---|---|---|---|---|
| Projects listed as funded on website | 5288 | 1164 | 173 | 104 | 6729 |
| Projects remaining after initial filter | 2019 * | 633 ** | 173 | 104 | 2929 |
| Projects receiving secondary screening | 53 | 11 | 10 | 5 | 79 |
| Projects meeting our search criteria | 8 ** | 1 | 1 | 0 | 10 |

* Project list was filtered using the facet search feature, filtering by the "Focal Area" of "Climate Change"; ** Project list was filtered using the "Biomes and ecosystem themes" feature, filtering by the "Mediterranean", "Island biodiversity", "Marine and Coastal biodiversity", "Marine", "Costal", and "Polar" categories.

*4.3. Literature and Funding Summary*

　　Despite the broadly published knowledge on invasive vertebrate impacts and the demonstrable benefits of invasive vertebrate eradication for native biodiversity, this systematic review of the literature and funded projects indicates that the potential for eradication of invasive vertebrates on islands to provide climate solutions is not fully realized. Indeed, we found few examples of eradications being proposed as a cross-sector conservation action and even fewer examples where the benefits of eradication as a climate solution were measured. However, we also acknowledge that our search terms did not capture all relevant studies (i.e., non-vertebrates). Further investigation of non-vertebrate eradications on climate adaptation and resilience, would likely reveal further support that eradication can achieve desired climate and biodiversity outcomes. As another example, de Wit et al. [70] highlighted several studies that demonstrate the importance of invasive vertebrate eradications for climate change-related sustainable development goals. These studies primarily relate to carbon sequestration, a search term that they explicitly used. By contrast, we did not include specific impacts or solutions in our searches, instead, we focused on studies that explicitly mention climate change in the context of invasive vertebrate eradications. Furthermore, our retrospective review of climate-funded projects by large global initiatives describes a minimum number of projects that explicitly target climate impacts. For example, it did not include current funding opportunities or those at a regional scale, such as for the Pacific Islands, where these types of funding proposals are currently initiated (SPREP 2021). Despite finding a limited number of studies and projects from our systematic reviews, it is clear that there is promising research at the intersection of IAS eradication and climate solutions [82,180]. The lack of eradication projects receiving funding from climate change funding sources is a lost opportunity to achieve cross-sector conservation successes.

**5. Conclusions**

　　IAS are one of the leading factors driving native biodiversity loss and species extinctions, with cascading impacts on human food security and livelihoods. The negative impacts of IAS on native biodiversity loss are compounded by climate change, which facilitates the spread and establishment of IAS, creates new opportunities for other species to become invasive and increases their negative impacts. In turn, the loss of native biodiversity from both IAS and climate change reduces ecosystem resilience to climate change impacts and increases vulnerability to the establishment of IAS. In combination, the synergistic interactions between IAS, climate change, and native biodiversity loss create a

positive feedback cycle reinforcing these interactions (Figure 1). These relationships are pronounced on islands, but there is growing evidence that IAS eradications can help to break the positive feedback cycle, improve biodiversity outcomes (i.e., recovery of floral and faunal communities and ecosystem function), and increase ecosystem resilience across both terrestrial and marine environments (Figure 1). As such, the prevention (biosecurity), control, and eradication of IAS on islands are key to the dual crises of climate change and native biodiversity loss, with eradications of invasive mammals on islands a clear and tractable example of holistic cross-sector opportunity. Our review of the literature and funded projects, however, indicates a missed opportunity to realize the full positive impacts of IAS eradications. Meanwhile, our literature review indicates that actions to improve resilience through IAS eradications, sometimes paired with other restoration actions, are achieving important conservation objectives, although a connection with climate change is not always explicit.

This review summarized the known linkages between IAS threats, IAS eradication, climate change, and ecosystem resilience, and identified that data gaps still exist. We encourage eradication practitioners and researchers to consider the synergies of invasive species and climate impacts when developing eradication strategies and work together to improve understanding of these linkages, particularly where IAS eradication could aid in climate mitigation strategies. We also encourage climate change funding entities to increase funding to IAS eradications on islands, particularly for small island developing states that often have limited resources to address both the biodiversity and climate change crises that many are already experiencing.

**Supplementary Materials:** The following are available online at https://www.mdpi.com/article/10.3390/cli9120172/s1, Table S1. Summary of systematic literature review results. Table S2. Summary of studies from systematic literature review that passed the secondary screening. Table S3. Summary list of funded projects that proposed, at least in part, to eradicate an invasive mammal from an island(s) with the justification aimed at improving resilience to the impacts of climate change. Table includes the funding entity: Global Environment Facility (GEF), Darwin Initiative (DI), Green Climate Fund (GCF), and the Wildlife Conservation Society's Climate Adaptation Fund (WCS), name of project, the year of project funding, project country, target species for eradication (including where the target species was not made explicit ("TBD"), and name of the project island/archipelago.

**Author Contributions:** Conceptualization, P.J.K., C.E.B., D.R.S., C.A.W., D.J.W., N.D.H.; methodology, P.J.K., C.E.B., D.R.S., C.A.W., D.J.W.; formal analysis, P.J.K., C.E.B.; investigation, P.J.K., C.E.B.; resources, NA; data curation, P.J.K., C.E.B., D.R.S., D.J.W.; writing—original draft preparation, P.J.K., C.E.B., D.R.S., C.A.W.; writing—review and editing, P.J.K., C.E.B., D.R.S., C.A.W., D.J.W., N.D.H.; visualization, P.J.K., C.E.B.; supervision, P.J.K.; project administration, P.J.K.; funding acquisition, NA. All authors have read and agreed to the published version of the manuscript.

**Funding:** This research received no external funding. CEB was supported by the Bertarelli Foundation as part of the Bertarelli Programme in Marine Science.

**Acknowledgments:** We would like to thank K. Campbell for leading a discussion that initiated this manuscript. Comments from two anonymous reviewers and T. Kittel and T. Schulz greatly improved this manuscript.

**Conflicts of Interest:** The authors declare no conflict of interest.

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
