# Peer review of "Do Invasive Mammal Eradications from Islands Support Climate Change Adaptation and Mitigation?"

_climate, doi:10.3390/cli9120172_

Round 1
Reviewer 1 Report
I thought this was an excellent and thought-provoking review/perspective. I really enjoyed reading it, and I think it could serve as a timely article given the need to consider the climate change - biodiversity nexus. I've very few comments, and these act as suggestions rather than requirements. Well done.
- I would move table 1 to the appendix. This journal allows appendices to be added after the references, and in my opinion this is the most suitable place for the table.
- I would include the methods section before the discussion stemming from it, as there is some redundant material in there. It might streamline the paper just a little bit more
- I think you've missed an opportunity for a figure to summarize the key findings from your analysis. A simple bar chart of n articles across your three topics can provide the readers/funders with a clearer picture of what was missing.
Minor
- figure captions below figures
- double spacing on L276
Reviewer 2 Report
The presented manuscript summarizes the knowledge about invasive alien species and their eradication in the context of climate change. Rightly so, the authors emphasized that invasive vertebrates (here focused on invasive mammals) are known to be ubiquitous, especially on islands, and therefore their eradication may play a role as an effective climate adaptation strategy. The authors, as they wrote, focused on two main topics i.e. the important characteristics that drive the positive feedback cycle between invasive alien species, climate change, biodiversity loss, and when eradications can likely contribute to improving resilience and adaptation to climate change, or serve to mitigate climate impacts, including case studies of the impacts of invasive alien species and climate change on terrestrial and near-shore marine ecosystems.
In my opinion, such an approach is indeed well thought out. I do not have any doubts about the manuscript organization. The sections are well prepared and described with sufficient literature and thoroughly searched examples of funding projects proposing to eradicate invasive vertebrates from islands.
The important part is the conclusion section. The authors stated that: Our review of the literature and funded projects, however, indicates a missed opportunity to realize the full positive impacts of IAS eradications. Meanwhile, our literature review indicates that actions to improve resilience through IAS eradications, sometimes paired with other restoration actions, is achieving important conservation targets, yet connectivity with climate change is not always explicit. Such a statement is crucial in understanding the presented problem.
After reading the article carefully, I found no shortcomings of a scientific or editorial nature.
To conclude, I think that the presented article summarizes the knowledge on a specific topic and gives sufficient conclusions for both scientists and practitioners.
It was a pleasure for me to read the manuscript. The article is well prepared. Congratulations!
